# Exposure Risk to Medical Staff in a Nasopharyngeal Swab Sampling Cabin under Four Different Ventilation Strategies

Jianchao Ma [1], Hua Qian [1,*], Fan Liu [1,2], Guodong Sui [3] and Xiaohong Zheng [1,4]

1 School of Energy and Environment, Southeast University, Nanjing 210096, China; jianchao_ma@seu.edu.cn (J.M.); 230179041@seu.edu.cn (F.L.); xhzheng@seu.edu.cn (X.Z.)
2 School of Environment and Architecture, University of Shanghai for Science and Technology, Shanghai 210093, China
3 Shanghai Key Laboratory of Atmospheric Particle Pollution and Prevention (LAP3), Department of Environmental Science and Engineering, Fudan University, Shanghai 200438, China; gsui@fudan.edu.cn
4 Jiangsu Provincial Key Laboratory of Solar Energy Science and Technology, School of Energy and Environment, Southeast University, Nanjing 210096, China
* Correspondence: qianh@seu.edu.cn

**Abstract:** Medical staff working in a nasopharyngeal swab sampling cabin are exposed to a higher exposure risk of COVID-19. In this study, computational fluid dynamics (CFD) are used to evaluate the exposure risk to medical staff in a nasopharyngeal swab sampling cabin of Chinese customs under four different ventilation strategies, i.e., multiple supply fans ventilation (MSFV), multiple exhaust fans ventilation (MEFV), single exhaust fan and outer windows closed ventilation (SEFV), and single exhaust fan and outer windows opened ventilation (SEFV-W). The impact of physical partitions on exposure risk is also discussed. The results show that MSFV performed best in reducing exposure risk. No significant difference was found between MEFV and SEFV. SEFV-W performed better than SEFV with a ventilation rate of 10–50 L/(s·Person), while it performed worse with a ventilation rate of 50–90 L/(s·Person). The exposure risk to medical staff did not decrease linearly with the increase in the ventilation flow rate under the four ventilation strategies. For MSFV, the installation of partitions is conducive to the reduction in the exposure risk. This study is expected to provide some guidance for ventilation designs in sampling cabins.

**Keywords:** ventilation strategy; COVID-19; ventilation rate; physical partition

## 1. Introduction

The ongoing novel coronavirus disease 2019 (COVID-19) pandemic, caused by the severe acute respiratory syndrome coronavirus 2 (SARS-CoV-2), has created a global health and financial crisis [1]. During the COVID-19 pandemic, medical staff who collect specimens in a nasopharyngeal swab sampling room have direct contact with infected patients, thus facing a higher infection risk [2,3]. Understanding how to mitigate the COVID-19 transmission risk using indoor ventilation strategies can deliver guidance for reasonable precautions to reasonably protect medical staff in such emergencies.

SARS-CoV-2 was considered to be transmitted via the virus-laden aerosols emitted by human respiratory activities [4,5], such as breathing [6], talking [7], coughing [8,9], and sneezing [10]. The aerosols of <5 μm may travel and be suspended for long distances, which is found to be largely determined by indoor airflow patterns and temperature distribution by full-scale experiments and numerical investigation [11–13]. Cho [14] evaluated the performance of three ventilation strategies on the protection of medical staff and the control of airborne infectious diseases in the negative pressure isolation room through numerical simulation and on-site measurement. These studies showed that increasing ventilation rate could reduce the cross-infection risk by removing or diluting pathogen-laden aerosols; however, some ventilation systems, such as displacement ventilation, could even increase

the cross-infection risk of respiratory diseases, especially via airborne transmission. Some recent studies have reiterated the association between insufficient ventilation and increased infection risk of COVID-19 [15–18]. Thus, effective ventilation is essential for minimizing the exposure risk to medical staff.

The effects of ventilation on the transmission of exhaled aerosols have been studied in some specified medical settings, such as general wards [11,19], consulting wards [20], isolation rooms [14,21], ICU rooms [22,23], and operation rooms [24,25]. During the COVID-19 pandemic, nasopharyngeal swab sampling is the primary test method for diagnosing SARS-CoV-2. Sampling cabins have been widely established in hospitals, customs, and communities so that the public can be sampled as conveniently as possible. The medical staff may be exposed to potential high-virus-load patients during the sampling process, such as asymptomatic and symptomatic infectors. However, less attention has been paid so far to explore how to reduce the exposure risk to medical personnel in the sampling process through effective ventilation.

In addition to ventilation, some physical protection measures, such as wearing masks [26], maintaining social distancing [27], and installing partitions [28,29], are recommended to mitigate the spread of infection. It has been identified that facemasks and social distancing can help reduce exposure levels to virus-laden aerosols. However, there may be some controversial opinions on the use of partitions. Some studies showed that the installation of partitions may increase the risk of exposure due to the interaction of indoor airflows, including ventilation airflow, human thermal plumes, and exhaled airflows, with partitions [30,31]. Therefore, it is of great significance to access the wide use in the blocking effect of the partition on the aerosol propagation path in the sampling cabin.

In this study, we focused on the spread of aerosols and the exposure risk to medical staff in the nasopharyngeal swab sampling cabin of Chinese customs. In the physical model, the sampling cabin is divided into a medical staff zone (MSZ) and a passenger zone (PZ) by physical partition. There are openings (inner windows) in the partition so that the medical staff sitting in the MSZ can take samples from the passengers sitting in the PZ. With the idea of preventing passenger-exhaled, viral-carrying aerosols from entering MSZ through the inner window, some ventilation strategies with different supply or exhaust positions have been adopted. For example, supply or exhaust fans may be installed in the medical staff or passenger areas. This study used computational fluid dynamics (CFD) to simulate the aerosol ($CO_2$ tracer gas) dispersion exhaled by the passenger in the nasopharyngeal swab sampling chamber under four typical ventilation strategies, with the impact of physical partitions considered. Dilution ratio ($\varepsilon$) at the respiratory height in the medical staff zone, the inhaled fraction (IF), and the infection risk (IR) to the medical staff are used to assess the exposure risk. This work is expected to provide scientific guidance for the sampling chamber's ventilation strategy and physical configuration design.

## 2. Materials and Methods

### 2.1. Physical Model

Figure 1 shows a plan view of a nasopharyngeal swab sampling cabin built in Chinese customs, where the medical staff collected nasopharyngeal swab samples from all passengers entering the customs. The size of the sampling chamber is 30 m (length (x)) × 6 m (width (z)) × 2.7 m (height (y)). The sampling cabin was divided into the medical staff zone (MSZ) and the passenger zone (PZ) by central partition, and 22 workstations were separated by lateral partitions in the sampling cabin. Each workstation had an inner window with a size of 0.8 m × 0.8 m (its low edge was 0.8 m from the ground) on the central partition to facilitate the medical staff in the MSZ to collect samples from the passengers in the PZ. Ten circular air supply openings with a diameter of 0.3 m were placed on the back wall in the MSZ to provide clean outdoor air to the sampling cabin. Twelve outer windows with a size of 0.8 m × 0.8 m were arranged on the exterior wall in the PZ, which can be opened and closed freely as needed. Four doors were installed on the outer wall and the central partition. All doors were closed during sampling and treated as walls or partitions when

building the physical model. To simplify the numerical simulation, symmetric boundary conditions were applied to the lateral boundaries (x = 15 m) and the computational domain can be reduced to a half-space.

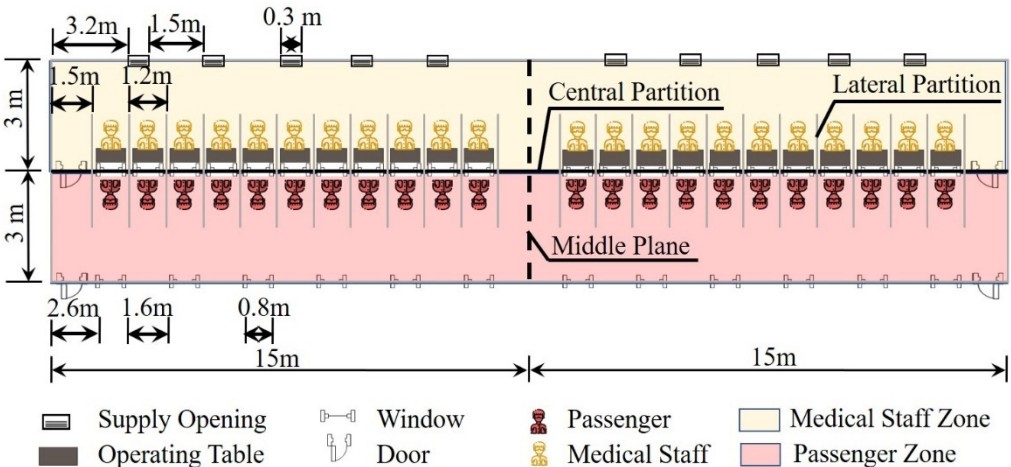

**Figure 1.** Plan view of a sampling cabin.

Figure 2 shows the coordinate system of the 3D physical model, as well as the positions of supply openings (S1–S5), outer windows (W1–W6), etc. The bodies of medical staff and passengers (P1 to P11) were from practical medical scans. They were sitting in chairs, face-to-face, with a distance of 1.15 m between their mouths, which represented a typical scene where passengers were waiting for specimen collection.

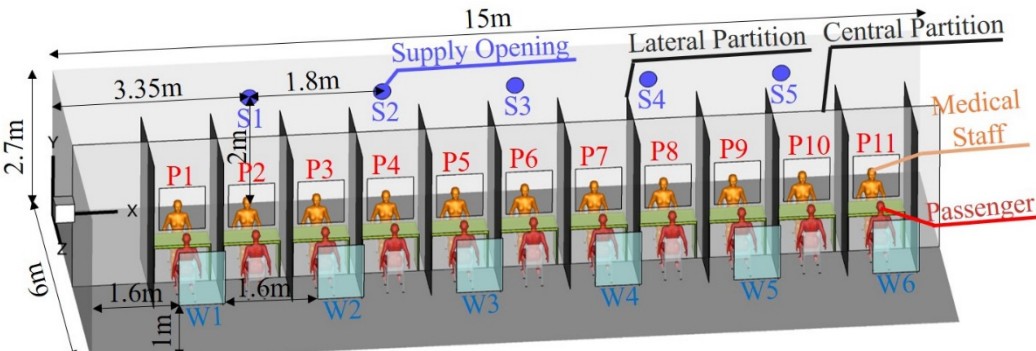

**Figure 2.** Three-dimensional physical model.

Four typical ventilation strategies were employed for comparison in this study, as shown in Figure 3. For multiple supply fans ventilation (MSFV), see Figure 3a, five identical air supply fans were installed at the air supply openings in the MSZ, and all outer windows (W1–W6) in PZ were opened. For multiple exhaust fans ventilation (MEFV), see Figure 3b, six identical exhaust fans were installed at the outer window in the PZ. For single exhaust fan ventilation (SEFV), see Figure 3c, only one exhaust fan was installed in the W1, and the rest of the windows were closed. In single exhaust fan and outer windows opened ventilation (SEFV-W), see Figure 3d, the ventilation configuration was the same as SEFV, except that the other windows (W2–W6) were opened.

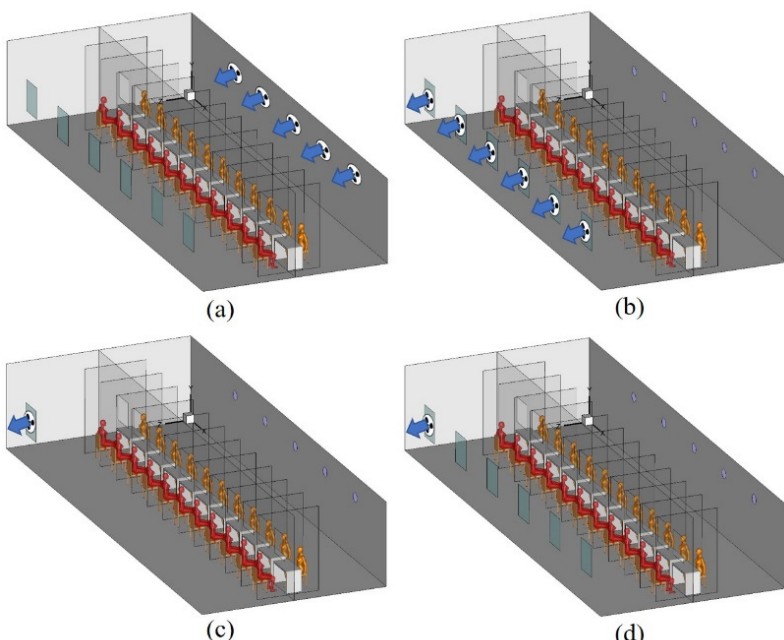

**Figure 3.** Ventilation strategies for a nasopharyngeal swab sampling cabin: (**a**) MSFV; (**b**) MEFV; (**c**) SEFV; (**d**) SEFV-W.

### 2.2. Boundary Conditions

The infection risk of indoor personnel may be affected by the number and location of the infectors [32]. The protective effect of ventilation on the medical staff in one of the most unfavorable situations was discussed by assuming that all passengers are infectors and all medical staff are susceptible in this study. The passengers exhaled the virus-laden aerosols through their mouth (with a hydraulic diameter of 0.016 m) with a stable temperature of 307 K at a constant velocity of 1 m/s (normal breathing), and the medical staff inhaled air with the same velocity. In the normal breathing process, most of the aerosol particles are smaller than 0.3 μm, and a few of them (<2%) are larger than 1 μm in diameter [33,34]. Small aerosols can suspend in the air and follow the airflow, so they can be simulated by the tracer gas [35]. The commonly used tracer gas $CO_2$ with mole fraction at 4% was adopted to simulate virus-laden airborne exhaled from the passengers [36]. There were no other heat sources in the sampling cabin with adiabatic walls, except for the medical staff and the passengers. The convection heat flux at body surfaces was defined as 30 W/m² for medical staff and passengers (with a surface area of 1.25 m²). Non-slip boundary conditions were applied for all walls.

The clean outdoor air temperature was 300 K, and it was delivered to the cabin from the supply opening with different ventilation flow rates (10, 20, 30, 50, 70, and 90 L/(s·Person)), which fulfilled the national minimum requirements set in the civil building code [37–39]. The detailed settings of the air supply openings and outer windows under different ventilation strategies are shown in Table 1.

### 2.3. Numerical Methods and Grid Independency

The airflow inside the cabin was assumed to be steady, three-dimensional, incompressible, and with a turbulent flow. The widely used standard k–ε model [40] was used to simulate turbulent flow. The enhanced wall treatment was adopted to treat the turbulent airflow properties in near-wall regions. A species transport model, based on the Eulerian method, was applied to calculate the dispersion process of the passengers' exhaled flow [41]. The governing equations were solved using ANSYS Fluent [42] and were discretized using second-order upwind schemes. The SIMPLE algorithm was adopted to decouple pressure and velocity [43]. The scaled residuals should be less than $10^{-4}$ for all quantities except

energy, where it should be less than $10^{-7}$ for convergence, and the net mass rates were less than 0.01% [44].

**Table 1.** The detailed settings of the air supply openings and outer windows under different ventilation strategies.

| Ventilation Strategy | Supply Openings | Outer Windows |
|---|---|---|
| MSFV | Velocity Inlet (10, 20, 30, 50, 70, 90 L/(s·Person)) | W1–W6: Pressure Outlet |
| MEFV | Pressure Outlet | W1–W6: Velocity Inlet (−10, −20, −30, −50, −70, −90 L/(s·Person)) |
| SEFV | Pressure Outlet | W1: Velocity Inlet (−10, −20, −30, −50, −70, −90 L/(s·Person)); W2–W6: Wall |
| SEFV-W | Pressure Outlet | W1: Velocity Inlet (−10, −20, −30, −50, −70, −90 L/(s·Person)); W2–W6: Pressure Outlet |

GAMBIT [45] was used for constructing the physical models and generating grids, and the computational domain meshed with tetrahedral elements. Refined boundary layers were employed near the human body, mouths, supply openings, outer windows, and inner windows, where the temperature and velocity may be higher. The mesh sizes of the mouth were set as 0.002 m, which were smaller than 0.02 m around the human body. The dimensionless wall distance $y^+$ was less than 5 at the near-wall node on the human body. Similarly, the mesh sizes on the supply openings, inner windows, and outer windows were 0.01 m. While for the computational domain, the maximum mesh sizes were 0.1 m.

The grid independence was performed by taking the MSFV as an example, in which the velocity profiles in two vertical lines at x = 7.5 m, z = 1.5 m and x = 7.5 m, z = 4.5 m under coarse (9.2 million cells), medium (11.4 million cells), and fine mesh (15.4 million cells) sizes were compared when the ventilation flow rate was 10 L/(s·Person). The results are shown in Figure 4. It can be found the velocity profile predicted at the lines by the coarse mesh and the fine mesh was quite different, with a maximum difference of 23% and 8%, respectively. In contrast, the results by the medium mesh and the fine mesh at the lines had good consistency, with the maximum difference being only 5% and 2%, respectively. The simulation results obtained by using middle mesh and fine mesh under other ventilation strategies also had a good consistency. Therefore, the medium mesh with 11.4 million cells was ultimately selected for the simulations.

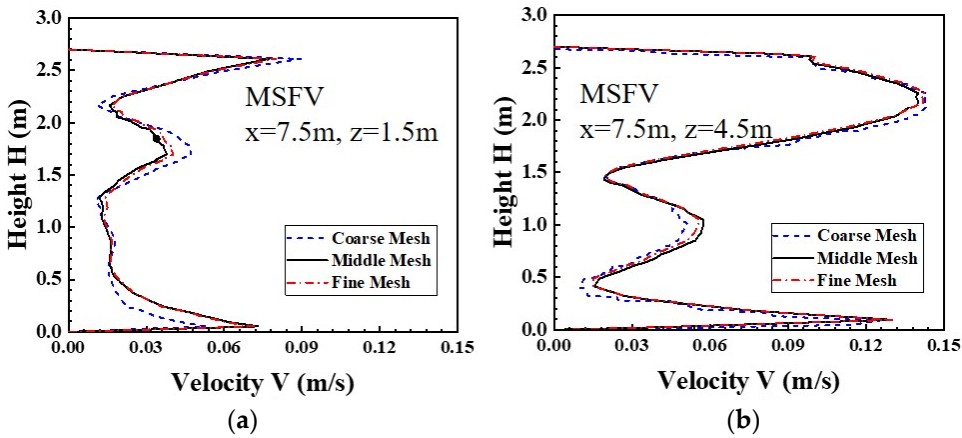

**Figure 4.** Velocity profiles in a vertical line at two lines under different mesh sizes. (**a**): x = 7.5 m, z = 1.5 m; (**b**): x = 7.5 m, z = 4.5 m.

### 2.4. Evaluation Indexes

The dilution ratio ($\varepsilon$) and intake fraction (*IF*) used in the literature [31] were adopted to access the exposure levels of the medical staff, which were defined by the formula shown below:

$$\varepsilon = \frac{C}{C_e} \tag{1}$$

$$IF = \frac{q_i}{q_e} = \frac{C_i}{C_e \times I} \tag{2}$$

where $C$ is the tracer gas concentration of a point somewhere in the cabin, $C_e$ is the exhaled tracer gas concentration of passenger, $q_i$ is the inhaled tracer gas flow rate of exposed personnel, $q_e$ is the total exhaled tracer gas flow rate of infectors, $C_i$ is the inhaled tracer gas concentration of exposed personnel, $C_e$ is the exhaled tracer gas concentration of infector, and $I$ is the number of infectors.

The infection risk, based on the Wells–Riley equation was another main evaluation index to accurately quantify the risk to the exposed personnel. The detail of the Wells–Riley equation is shown as follows [46]:

$$P = 1 - e^{-Iqpt/Q} \tag{3}$$

To include the impact of the non-uniform distribution of indoor aerosols induced by the locations of the infectious and the susceptible, and the airflow pattern, in this work, a modified Wells–Riley model adopted in studies [31,47] is used to calculate the spatial distribution of infection risk, expressed as follows:

$$P = 1 - e^{-qC_it/C_e} = 1 - e^{-Iqt \times IF} \tag{4}$$

where $P$ is the infection probability, $q$ is the quantum generation rate by an infector (quanta/h), $p$ is the pulmonary ventilation rate (m$^3$/h), $t$ is the total exposure time (h), and $Q$ is the room ventilation rate (m$^3$/h). Taking the effect of wearing masks into account, the inhaled tracer gas concentration of exposed personnel is $C_i \times (1 - \eta_i) \times (1 - \eta_e)$, and the infection risk can finally be calculated as:

$$P = 1 - e^{-Iqt \times IF \times (1-\eta_i) \times (1-\eta_e)} \tag{5}$$

where $\eta_i$ and $\eta_e$ are mask efficiencies for the infectors and exposure persons, respectively. All medical staff were wearing N95 masks throughout the sampling period, while passengers took off masks, and waited for sampling. Therefore, the mask efficiencies for the medical staff and passengers were $\eta_i = 90\%$ and $\eta_e = 0$, respectively [48]. To assess the maximum risk level under adverse conditions, a high $q$ value (48 quanta/h) [49] and a long exposure time $t$ (3 min) [50] were selected in the following simulations.

### 3. Model Validation

Two benchmark cases from the literature were simulated to validate the CFD model. Case 1 was about forced convection in a two-zone chamber, which was measured by Posner [51]. Case 2 was designed to study the indoor air and tracer gas distribution in a two-person office with displacement ventilation, which was measured by Yuan [52]. Detailed model validation can be found in Supplementary Materials. The comparison between the simulation results and the experimental results for Case 1 and Case 2 were shown in Figures S1 and S2, respectively.

### 4. Results

#### 4.1. Airflow Patterns in the Sampling Cabin

The airflow pattern in the sampling cabin is fundamental for assessing the exposure risk to medical staff. Figure 5 shows the airflow patterns in the sampling cabin with

different ventilation flow rates under the MSFV. Three subgraphs under each ventilation rate, respectively, represent the velocity distribution at the breathing height (y = 1.1 m) (left subgraph), the velocity distribution at the S2 center section A-A' (x = 3.2 m), the streamlines (purple lines) of the exhalation flow of P2 (upper right subgraph), the velocity distribution at the P1 center section B-B' (x = 1.9 m), and the streamlines (purple lines) of the exhalation flow of P1 (lower right subgraph).

When the ventilation rate was 10 L/(s·Person), as shown in Figure 5a, the supply air jet from supply openings bent downwards due to the negative buoyancy and descended to the height of the inner window (upper right subgraph), then entered the PZ, which blow the exhaled airflow of P2 facing the supply opening S1 toward the outer window. Due to the "stack effect", caused by the air density difference between the indoor and outdoor, the outdoor air was induced to enter the PZ from the lower part of the outer window (lower right subgraph), and then impinged the back of P2. The exhaled airflow of P2 was taken away by the rising thermal plume generated by the human body and finally discharged outside from the upper part of the outer window. This indicated that the thermal plume generated by passengers may play a positive role in preventing exhaled pollutants from entering the MSZ. As the ventilation rate increased, the supply air jet gradually increased, while the "stack effect" decreases. For example, when the ventilation rate increased to 20 L/(s·Person), see Figure 5b, the supply air jet directly hit the intermediate partition, and then a part of the air flowed to the ceiling and returned to the MSZ, and the other stream of the air flowed downwards and entered the PZ from the inner window, and prevented the P1 exhaled polluted air from entering the MSZ. However, the airflow entering from the outer window became weaker, and its trajectory was lifted toward the head of P2, which caused the thermal plume to entrain the exhaled polluted air to flow toward the MSZ. This may increase the exposure risk to medical staff. When the ventilation rate increased to 30 L/(s·Person), see Figure 5c, the velocity of the air entering from the outer window continued to decrease. Although the airflow can still blow to the passenger's head, it has a weaker effect on the upward human plume. As the ventilation rate continued to increase from 50 L/(s·Person) to 90 L/(s·Person), see Figure 5d–f, the flow velocity in the sampling chamber increased, causing the backflow to disappear. This may be beneficial to quickly dilute and remove passengers' exhaled pollutants from the sampling room to reduce the exposure risk to medical staff.

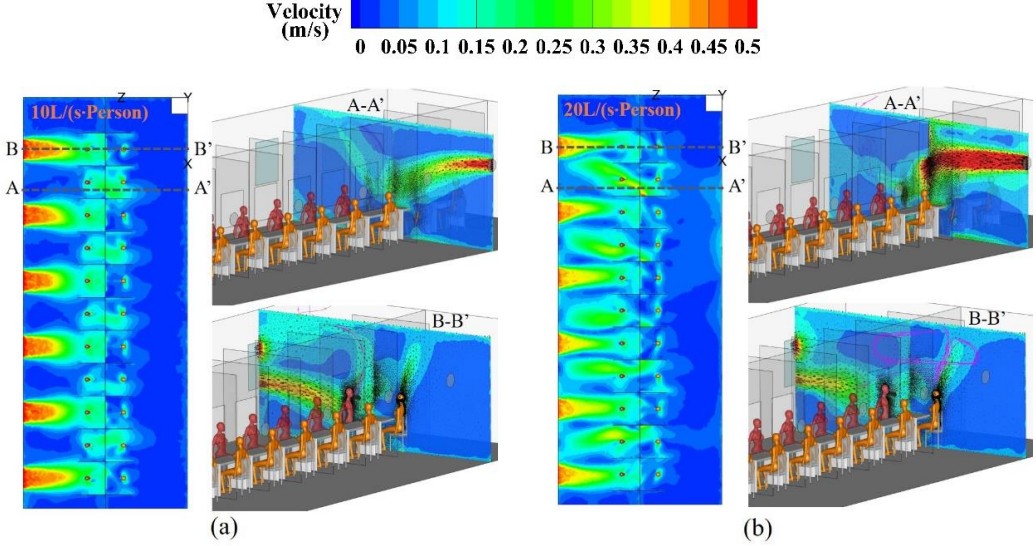

**Figure 5.** *Cont.*

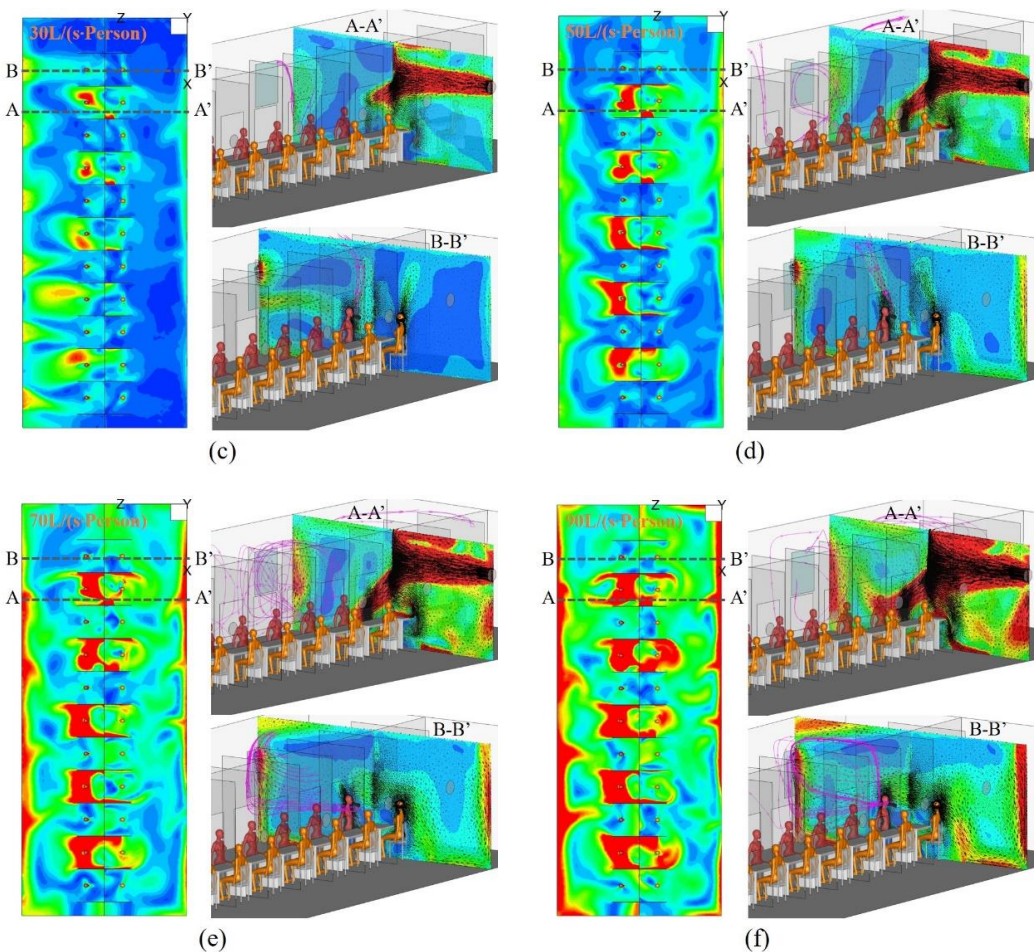

**Figure 5.** Airflow patterns at the planes of y = 1.1 m, x = 1.09 m, and x = 2.25 m under the MSFV with different ventilation rates: (**a**) 10 L/(s·Person); (**b**) 20 L/(s·Person); (**c**) 30 L/(s·Person); (**d**) 50 L/(s·Person); (**e**) 70 L/(s·Person); (**f**) 90 L/(s·Person).

Figure 6 shows the airflow patterns in the sampling cabin at different ventilation flow rates under MEFV. Comparing MSFV and MEFV, the former had a higher jet velocity at the supply openings. This may be due to the fact that the indoor air temperature discharged was higher than that of the outdoor fresh air. Although the ventilation rate delivered by the supply fans (in MSFV) and the exhaust fans (in MEFV) was the same, MSFV provided more fresh air flow from supply openings.

When the ventilation rate was 10 L/(s·Person) (see Figure 6a), the supply air jet from supply openings bent downward and hit the back of the medical staff, then bypassed the human body and entered the PZ. Since the MEFV used forced exhaust from the external windows, there is no airflow entering from the outer windows. When the ventilation rate increased to 20 L/(s·Person), the supply air jet directly entered the PZ through the inner window and blew the P2 exhaled pollutants to the outer window. Due to the jet entraining a large amount of ambient air to the PZ and cannot be discharged from the outer window in time, a part of the air returns to the MSZ from the adjacent inner window (lower right subgraph). Meanwhile, the pollutants exhaled by P1 will be carried to MSZ by the backflow, which may increase the exposure risk to medical staff. The supply air jet was strengthened as the ventilation flow rate increased, as shown in Figure 6c–e, where the jet directly impinged the center partition, and then was divided into two air streams. One stream entered the PZ from the inner window and the other airstream returned to the MSZ and induce a clockwise flow at the B-B' section (lower right subgraph), which will help prevent the contaminants exhaled by P1 from entering the MSZ. When the ventilation

rate increased to 70 L/(s·Person), the clockwise flow was enhanced, which strengthened the diffusion of P1 exhaled pollutants, potentially increasing the exposure risk. When the ventilation rate reached 90 L/(s·Person), the flow field distribution was similar to that with 70 L/(s·Person); however, the air velocity in the sampling room was significantly increased, which may be conducive to preventing pollutants from entering the MSZ and diluting the pollutants.

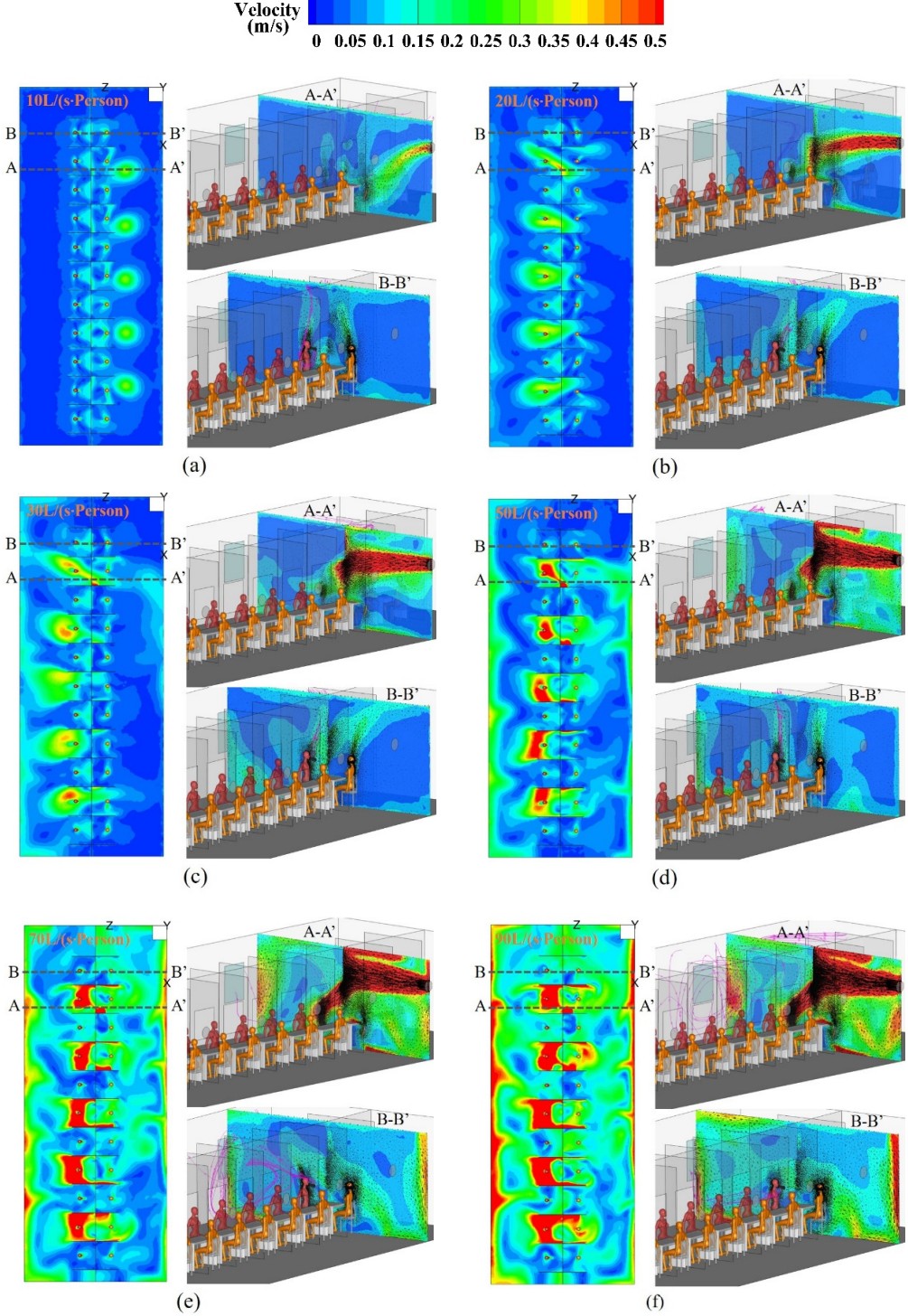

**Figure 6.** Airflow patterns at the planes of y = 1.1 m, x = 1.09 m, and x = 2.25 m under the MEFV with different ventilation rates: (**a**) 10 L/(s·Person); (**b**) 20 L/(s·Person); (**c**) 30 L/(s·Person); (**d**) 50 L/(s·Person); (**e**) 70 L/(s·Person); (**f**) 90 L/(s·Person).

Figure 7 shows the airflow patterns in the sampling cabin at different ventilation flow rates under SEFV. The airflow patterns in the sampling cabin under the two strategies of SEFV and MEFV are similar with a few differences. For example, when the ventilation rate was 20 L/(s·Person), as shown in Figure 7b, the backflow seemed to be weaker in the PZ under SEFV (lower right subgraph) and had a lower impact on the human thermal plume, so there may be less exhaled pollutants entering the MSZ. In addition, when the ventilation rate was 70 L/(s·Person), as shown in Figure 7e, the air volume discharged from W1 is greater under SEFV, and the exhaled pollutants of P2 can be quickly discharged, and the exhaled pollutants did not spread laterally as in the MEFV.

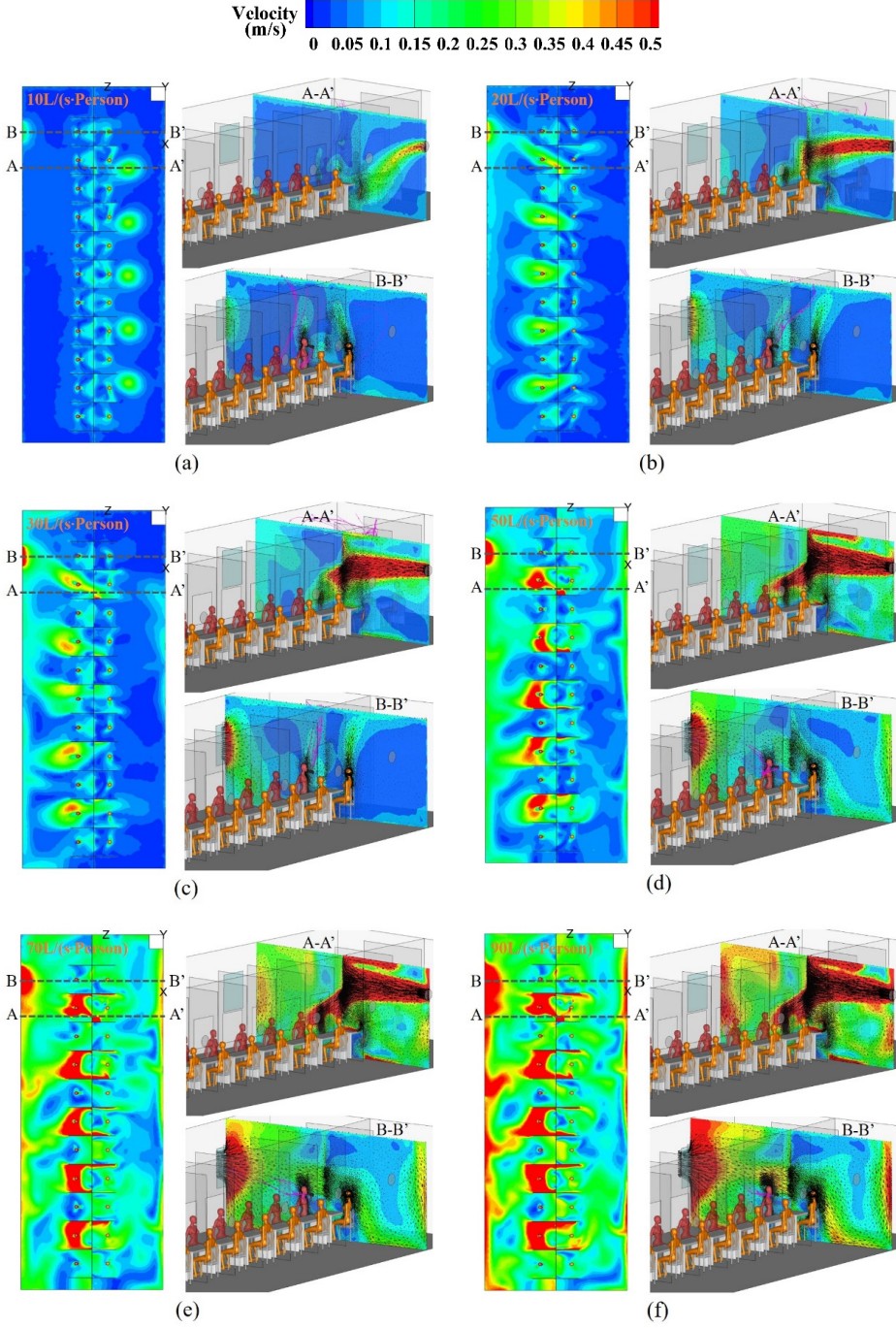

**Figure 7.** Airflow patterns at the planes of y = 1.1 m, x = 1.09 m, and x = 2.25 m under the SEFV with different ventilation rates: (**a**) 10 L/(s·Person); (**b**) 20 L/(s·Person); (**c**) 30 L/(s·Person); (**d**) 50 L/(s·Person); (**e**) 70 L/(s·Person); (**f**) 90 L/(s·Person).

Figure 8 shows the airflow patterns in the sampling cabin at different ventilation flow rates under SEFV-W. When the ventilation rate was 10 L/(s·Person), as shown in Figure 7a, the outdoor fresh air mainly entered the sampling room from the outer window due to the difference in air density, and less from supply openings. The fresh air entered from the outer window first enters the PZ and then passed through the inner window to reach the MSZ. The airflow direction was from the contaminated area to the clean area, going against the airborne transmission controls. Increasing the ventilation rate had a tiny effect on the airflow patterns, which may be caused by the air density difference dominating the indoor air movement.

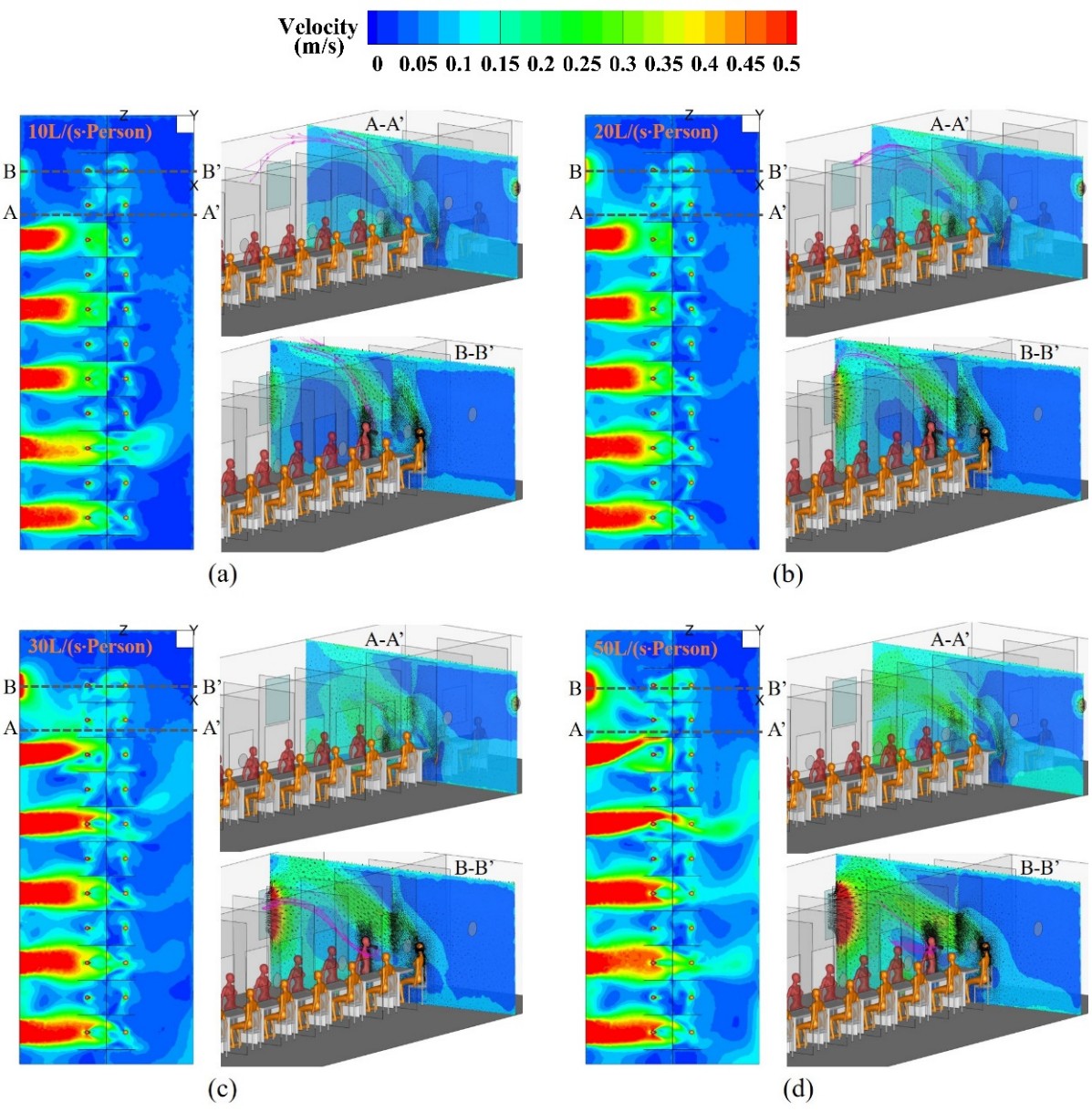

**Figure 8.** *Cont.*

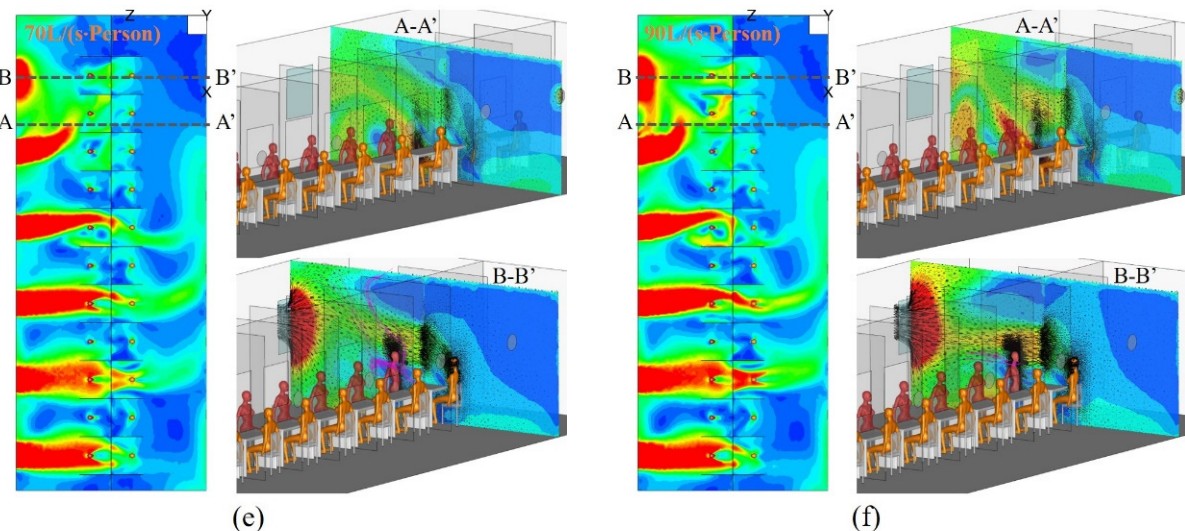

(e)                                                                (f)

**Figure 8.** Airflow patterns at the planes of y = 1.1 m, x = 1.09 m, and x = 2.25 m under the SEFV-W with different ventilation rates: (**a**) 10 L/(s·Person); (**b**) 20 L/(s·Person); (**c**) 30 L/(s·Person); (**d**) 50 L/(s·Person); (**e**) 70 L/(s·Person); (**f**) 90 L/(s·Person).

### 4.2. Dilution Ratio (ε) at the Breathing Height in the MSZ

Figure 9 shows significant differences in the average $\varepsilon$ under different ventilation strategies, and implies that increasing ventilation rate did not always reduce the exposure level of the medical staff.

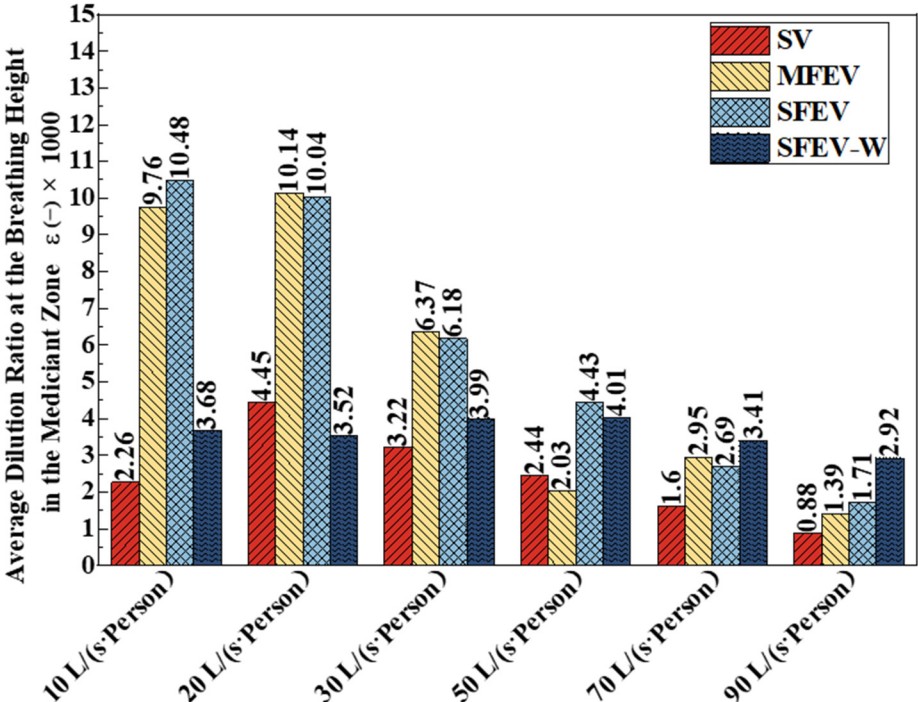

**Figure 9.** Average dilution ratio ($\varepsilon$) at the breathing height y = 1.1 m in the MSZ under different ventilation strategies with different ventilation rates.

For the MSFV, the detailed $\varepsilon$ distribution at breathing height is shown in Figure 10. Similar to Figure 9, the $\varepsilon$ at the respiratory height of MSZ first increased and then decreased. When the ventilation rate was 10 L/(s·Person), the $\varepsilon$ was $2.26 \times 10^{-3}$. As the ventilation rate increased to 20 L/(s·Person), the $\varepsilon$ did not decrease while peaking at $4.45 \times 10^{-3}$, and then $\varepsilon$ decreased with the ventilation rate. It may be because the airflow entering from

the outer window destroyed the thermal plume, and blew the exhaled pollutants from the passengers into the MSZ, as shown in Figure 5b.

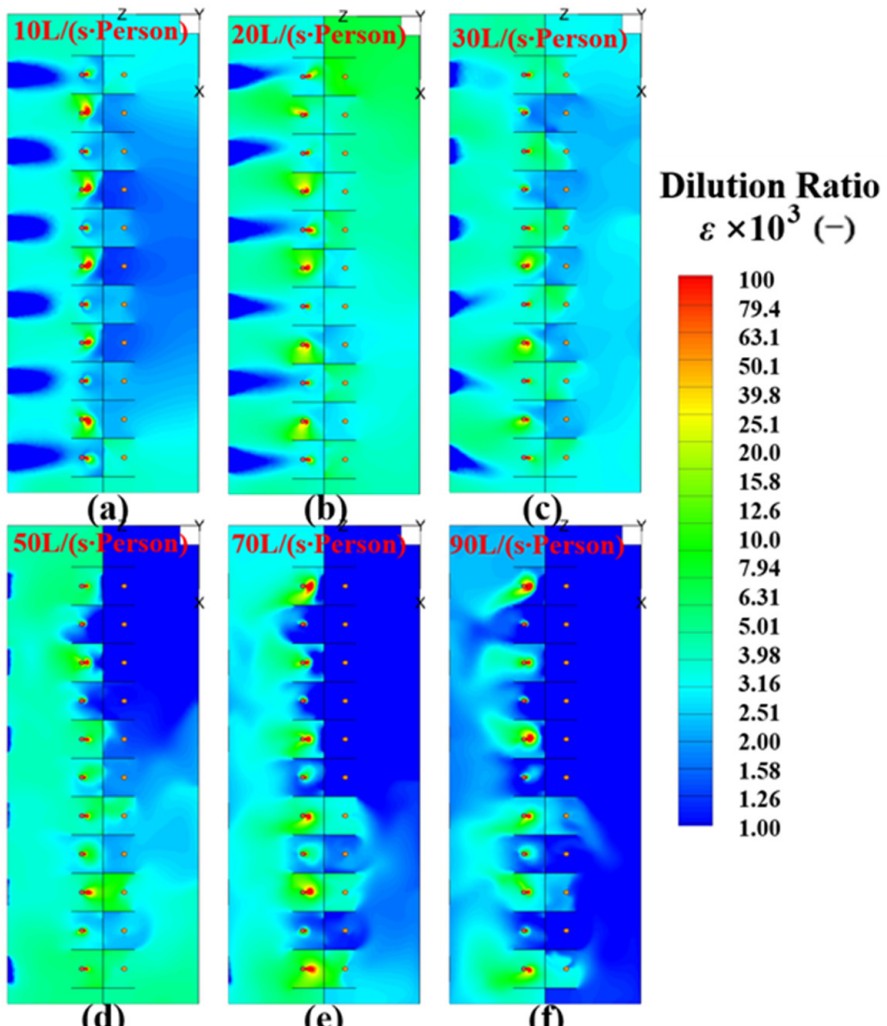

**Figure 10.** Dilution ratio distribution at the breathing height (y = 1.1 m) under the MSFV with different ventilation rates: (**a**) 10 L/(s·Person); (**b**) 20 L/(s·Person); (**c**) 30 L/(s·Person); (**d**) 50 L/(s·Person); (**e**) 70 L/(s·Person); (**f**) 90 L/(s·Person).

For MEFV, the detailed $\varepsilon$ distribution fluctuates as the ventilation rate increases, as shown in Figure 11, same with that in Figure 9. The $\varepsilon$ at the breathing height in the MSZ was $9.7 \times 10^{-3}$ with 10 L/(s·Person) and increased to $10.04 \times 10^{-3}$ with 20 L/(s·Person). However, the $\varepsilon$ was significantly reduced to $2.03 \times 10^{-3}$ with 50 L/(s·Person), while increasing again as the ventilation rate was increased to 70 L/(s·Person). After the ventilation rate increased to 90 L/(s·Person), the $\varepsilon$ dropped again. That means, the $\varepsilon$ peaked when the ventilation flow rate was 20 L/(s·Person) and 70 L/(s·Person). When the ventilation rate was 20 L/(s·Person), the supply air jet entrained a large amount of air and created a backflow in the PZ zone, which may bring the exhaled pollutants back to the MSZ. When the ventilation rate was 70 L/(s·Person), the clockwise airflow induced by the supply air jet was strengthened. The airflow entered the PZ and blew towards the passengers, strengthening the lateral dispersion of exhaled pollutants, further increasing the probability of pollutants reaching the MSZ.

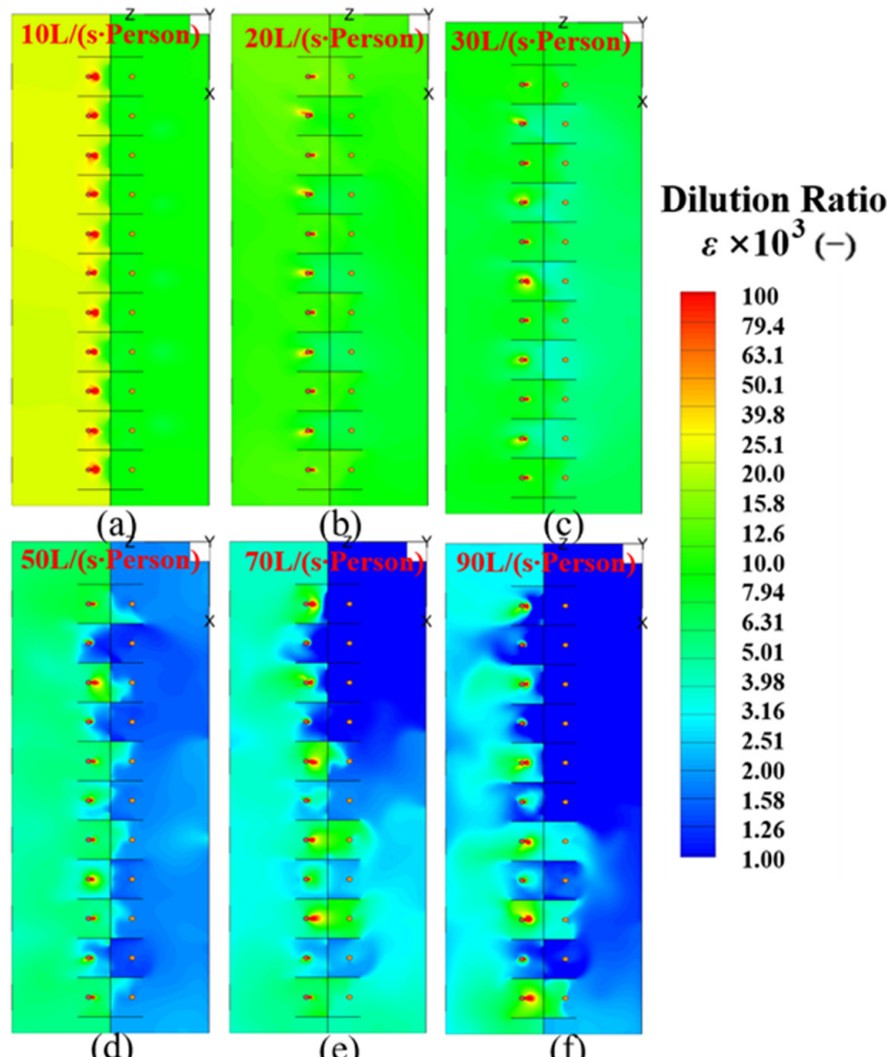

**Figure 11.** Dilution ratio distribution at the breathing height (y = 1.1 m) under the MEFV with different ventilation rates: (**a**) 10 L/(s·Person); (**b**) 20 L/(s·Person); (**c**) 30 L/(s·Person); (**d**) 50 L/(s·Person); (**e**) 70 L/(s·Person); (**f**) 90 L/(s·Person).

For SEFV, the detailed $\varepsilon$ distribution at breathing height is shown in Figure 12. It can be found from Figures 9 and 12 that the dilution ratio gradually decreased with the increase in the ventilation rate. The $\varepsilon$ distribution in the sampling cabin in SEFV is similar to that in MEFV, but with a little difference. For example, when the ventilation rate increased from 50 L/(s·Person) to 70 L/(s·Person), the $\varepsilon$ did not increase as in MEFV. This may be because the exhaust air volume of W1 was more in SEFV than in MEFV, which effectively reduced the dispersion of exhaled $CO_2$.

For SEFV-W, the detailed $\varepsilon$ distribution in Figure 13 did not change significantly with the increase in ventilation rate, similar to Figure 9. When the ventilation rate increased from 10 L/(s·Person) to 50 L/(s·Person), the $\varepsilon$ fluctuated around $4 \times 10^{-3}$; however, as the ventilation rate increased to 90 L/(s·Person), the dilution ratio dropped to $2.5 \times 10^{-3}$. It can also be seen from Figure 13 that the $\varepsilon$ in the PZ was lower than that in the MSZ. This may be because fresh air entered the sampling room mainly from the outer window (W2–W6), and the airflow rate was mainly affected by the temperature difference between indoor and outdoor air.

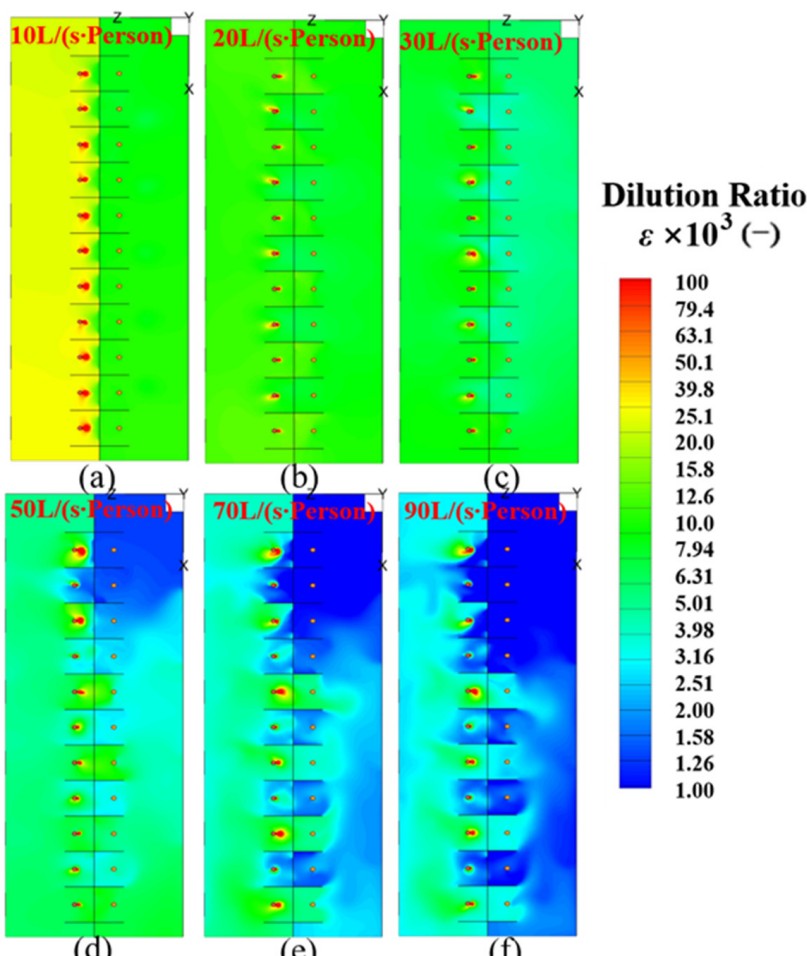

**Figure 12.** Dilution ratio distribution at the breathing height (y = 1.1 m) under the SEFV with different ventilation rates: (**a**) 10 L/(s·Person); (**b**) 20 L/(s·Person); (**c**) 30 L/(s·Person); (**d**) 50 L/(s·Person); (**e**) 70 L/(s·Person); (**f**) 90 L/(s·Person).

Increasing ventilation rate may raise the risk of exposure under specified conditions. Similar phenomena were also found in our studies. Generally, increasing the ventilation rate affects the spread of exhaled pollutions from the following three aspects: enhanced dilution, intensified dispersion, and the modified flow interaction. Thus, the influence of the airflow interaction on the airflow patterns should be carefully considered according to the specific situation while designing the ventilation rate.

Comparing the $\varepsilon$ at the breathing height in the MSZ from Figures 11–15, it can be found that the performance of the MSFV may be the best. This may be because more fresh air was provided from the supply openings in MSFV, better than the exhaust ventilation mode used in MEFV. The $\varepsilon$ in MEFV and SEFV were equivalent, indicating that the number of exhaust fans in this study had a negligible influence on the exposure level. Comparing SEFV and SEFV-W, when the ventilation rate was less than 50 L/(s·Person), the ventilation performance of SEFV-W was better, but when the ventilation rate was greater than that, SEFV was better. If single-window exhaust ventilation was adopted, then it was recommended to open doors and outer windows when the ventilation rate supplied by the exhaust fan was small, otherwise, close them when the ventilation rate was large. Therefore, the MSFV is recommended for adoption in the sampling cabin.

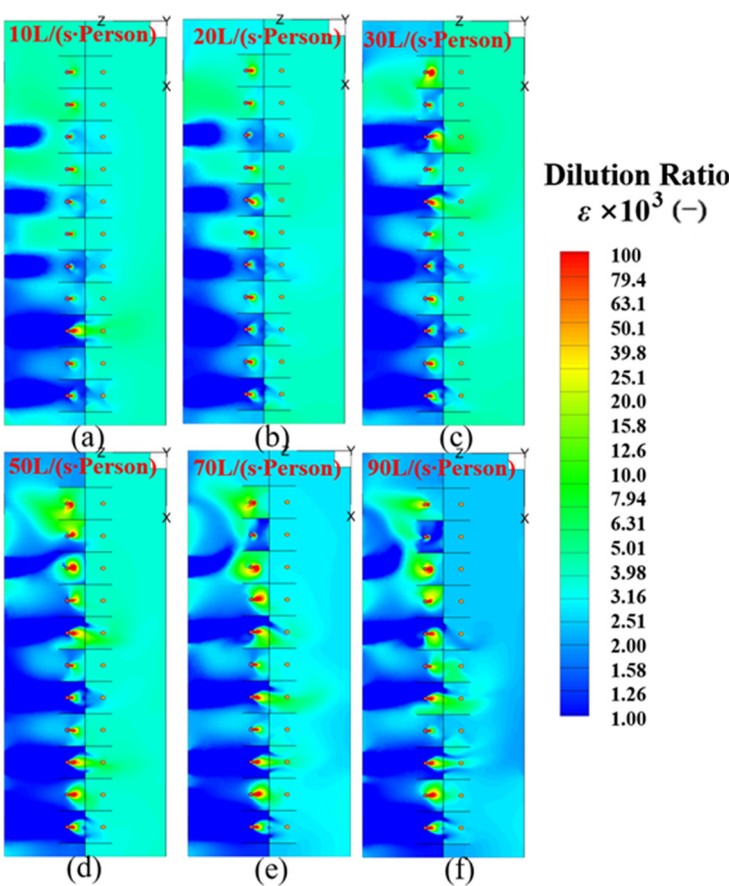

**Figure 13.** Dilution ratio distribution at the breathing height (y = 1.1 m) under the SEFV-W with different ventilation rates: (**a**) 10 L/(s·Person); (**b**) 20 L/(s·Person); (**c**) 30 L/(s·Person); (**d**) 50 L/(s·Person); (**e**) 70 L/(s·Person); (**f**) 90 L/(s·Person).

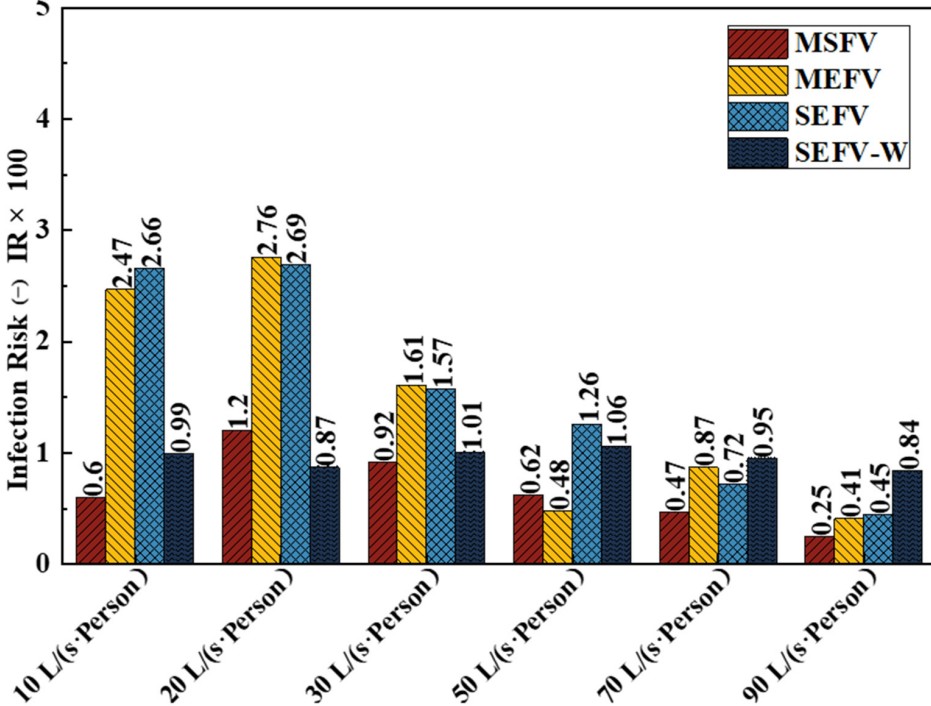

**Figure 14.** Infection risk to medical staff wearing N95 masks under the four ventilation strategies with different ventilation rates.

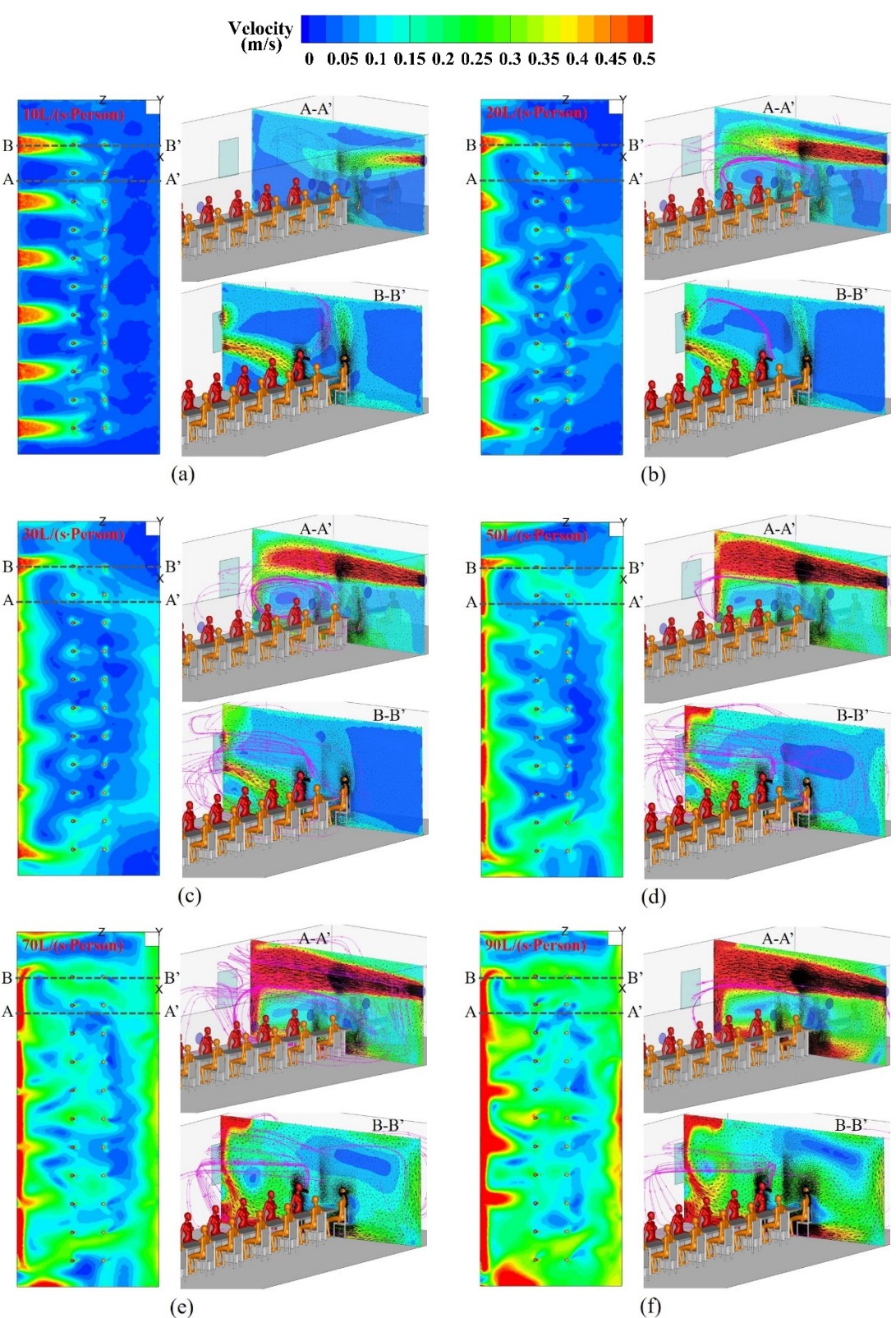

**Figure 15.** Airflow patterns at the planes of y = 1.1 m, x = 1.09 m, and x = 2.25 m under the MSFV without partition and with different ventilation rates: (**a**) 10 L/(s·Person); (**b**) 20 L/(s·Person); (**c**) 30 L/(s·Person); (**d**) 50 L/(s·Person); (**e**) 70 L/(s·Person); (**f**) 90 L/(s·Person).

### 4.3. Inhaled Fraction (IF) and Infection Risk (IR) to the Medical Staff

The *IF* and IR of medical staff can further quantify the impact of ventilation strategies on the exposure risk to medical staff. It was assumed that the medical staff wear N95 masks when collecting samples and filtration efficiency is 90% [48,53].

The *IF* values of the medical staff under the four ventilation strategies with different flow rates are listed in Table 2, from which we can see that the *IF* and the average $\varepsilon$ at the breathing height have almost the same trends as the ventilation flow rate increases. In addition, it can also be found that if the N95 masks were not worn, the amount of inhaled $CO_2$ accounts for 0.1–0.23%, 0.16–1.05%, 0.17–1.02%, and 0.32–0.40% of the exhaled $CO_2$, in the MSFV, MEFV, SEFV, and SEFV-W, respectively. If they wear N95 masks, the amount of inhaled $CO_2$ accounts is significantly reduced by 90%.

**Table 2.** Inhaled fraction of medical staff under different ventilation strategies $IF \times 100\%$.

| Flow Rate | MSFV | | MEFV | | SEFV | | SEFV-W | |
|---|---|---|---|---|---|---|---|---|
| L/(s·Person) | Without Mask | With Mask | Without Mask | With Mask | Without Mask | With Mask | Without Mask | With Mask |
| 10 | 0.23% | 0.02% | 0.94% | 0.09% | 1.01% | 0.10% | 0.38% | 0.04% |
| 20 | 0.46% | 0.05% | 1.05% | 0.10% | 1.02% | 0.10% | 0.33% | 0.03% |
| 30 | 0.35% | 0.03% | 0.61% | 0.06% | 0.59% | 0.06% | 0.38% | 0.04% |
| 50 | 0.23% | 0.02% | 0.18% | 0.02% | 0.48% | 0.05% | 0.40% | 0.04% |
| 70 | 0.18% | 0.02% | 0.33% | 0.03% | 0.27% | 0.03% | 0.36% | 0.04% |
| 90 | 0.10% | 0.01% | 0.16% | 0.02% | 0.17% | 0.02% | 0.32% | 0.03% |

The IR of the medical staff in the cabin could be calculated with Equation (5). Figure 16 shows infection risk to medical staff wearing N95 masks under the four ventilation strategies with different flow rates. According to Equation (5), the infection risk was positively correlated with the *IF*, so it was reasonable that the IR, the *IF*, and the average nominalized concentration at the breathing height kept almost the same trends. Overall, the risk of infection for medical staff under the MSFV was lower than that under the other three ventilation strategies. Therefore, the MSFV was recommended to be used in the sampling room to reduce the risk of infection.

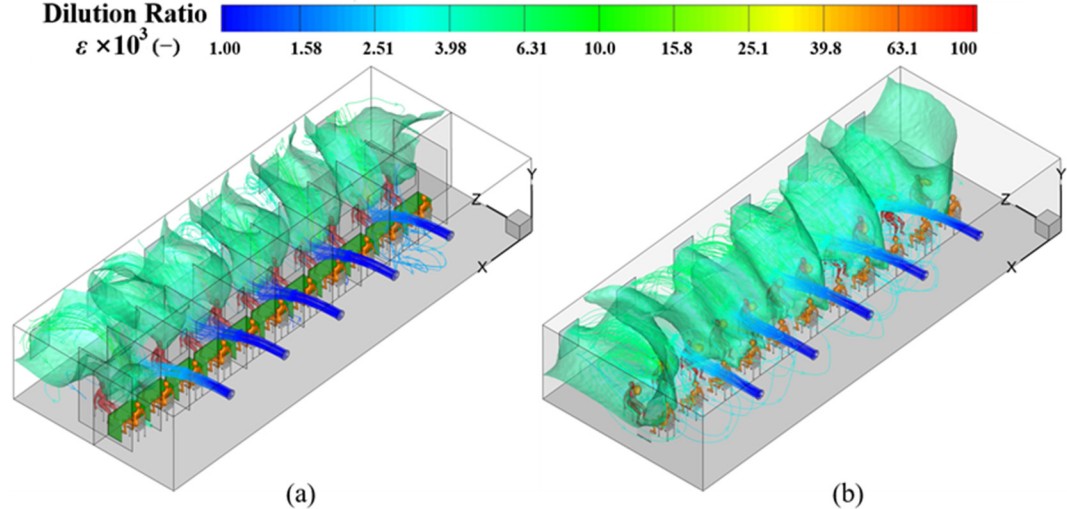

**Figure 16.** The streamline originating from the supply opening is colored by the dilution ratio, and the iso-surface with a dilution ratio of $5 \times 10^{-3}$: (**a**) before the partition was installed; (**b**) after the partition was installed.

It should be noted that the impact of ventilation rate on the IR was not straightforward in the MSFV. For example, the infection risk increased from 0.6% to 1.2% when the air

volume increased from 10 L/(s·Person) to 20 L/(s·Person), and when the ventilation rate increased from 20 L/(s·Person) to 90 L/(s·Person), the IR decreased from 1.2% to 0.25%. In this study, the IR was minimized when the ventilation flow rate reached 90 L/(s·Person). However, under this ventilation rate, the ventilation system may cause excessive energy consumption. In contrast, when the ventilation rate was 10 L/(s·Person), the IR was equivalent to the ventilation rate of 50 L/(s·Person). Therefore, from the perspective of reducing the risk of infection and reducing ventilation energy consumption, the ventilation rate of 10 L/(s·Person) was appropriate for the MSFV.

### 4.4. Influence of the Partitions on the Exposure Risk to Medical Staff

The *IF* and IR of the medical staff with or without partitions are compared in the sampling cabin under the MSFV, with the results shown in Table 3. The protective effect of masks was remarkable. The *IF* did not decrease monotonously as the ventilation rate increased. For example, increasing the ventilation rate from 10 L/(s·Person) to 20 L/(s·Person) and from 50 L/(s·Person) to 90 L/(s·Person) reduced the *IF* from 0.49% to 0.45% and from 0.57% to 0.40%, respectively; however, if medical staff did not wear the N95 masks, then increasing the ventilation rate from 20 L/(s·Person) to 50 L/(s·Person) caused the IR to increase from 0.45% to 0.57%.

**Table 3.** Inhaled fraction of the medical staff under the MSFV with and without partitions $IF \times 100\%$.

| Flow Rate L/(s·Person) | MSFV with Partition | | MSFV without Partition | |
|:---:|:---:|:---:|:---:|:---:|
| | **Without Mask** | **With Mask** | **Without Mask** | **With Mask** |
| 10 | 0.23% | 0.02% | 0.49% | 0.05% |
| 20 | 0.46% | 0.05% | 0.45% | 0.04% |
| 30 | 0.35% | 0.03% | 0.50% | 0.05% |
| 50 | 0.23% | 0.02% | 0.57% | 0.06% |
| 70 | 0.18% | 0.02% | 0.40% | 0.04% |
| 90 | 0.10% | 0.01% | 0.40% | 0.04% |

The changing trends of the *IF* may be caused by the different airflow distribution under different ventilation rates. Figure 15 shows the airflow patterns at the planes of y = 1.1 m, x = 1.09 m, and x = 2.25 m under the MSFV with different ventilation rates and without partition. As shown in Figure 15a, when the ventilation rate was 10 L/(s·Person), the supply air jet blew over the head of the passengers while bringing the passengers exhaled $CO_2$ to the outer windows. When the ventilation rate was 20 L/(s·Person), the supply air jet induced a counterclockwise vortex, which may cause $CO_2$ retention. This can explain why the ventilation rate increased from 10 L/(s·Person) to 20 L/(s·Person), but the *IF* was only slightly reduced by 0.04%. As the ventilation rate increased, the vortex intensity continued to increase, and the air velocity at the breathing height increased, which may enhance the diffusion of $CO_2$ exhaled by the passengers and lead to an increase in the *IF*. When the ventilation rate increased above 50 L/(s·Person), the dilution effect of the ventilation airflow dominated, leading to a reduction in *IF*.

Table 3 compares the *IF* of the medical staff with and without partition in the MSFV and shows that the partition can reduce the exposure risk to medical staff in most cases. This may be because the air velocity of the medical staff was higher and no vortex occurs occurred with a partition installed, as shown in Figures 5 and 15. Taking the ventilation rate of 10 L/(s·Person) as an example, the streamline originating from the supply opening colored by the normalized concentration, and the iso-surface with a nominal concentration of $5 \times 10^{-3}$ is shown in Figure 16. It can be found that the area affected by high concentration ($\varepsilon > 0.005$) was reduced after dividers were installed. It should be noted that when the ventilation flow was 20 L/(s·Person), regardless of whether the partition was installed, the *IF* of the medical staff was equivalent. Similar conclusions can be drawn from the IR of the medical staff, as shown in Figure 17.

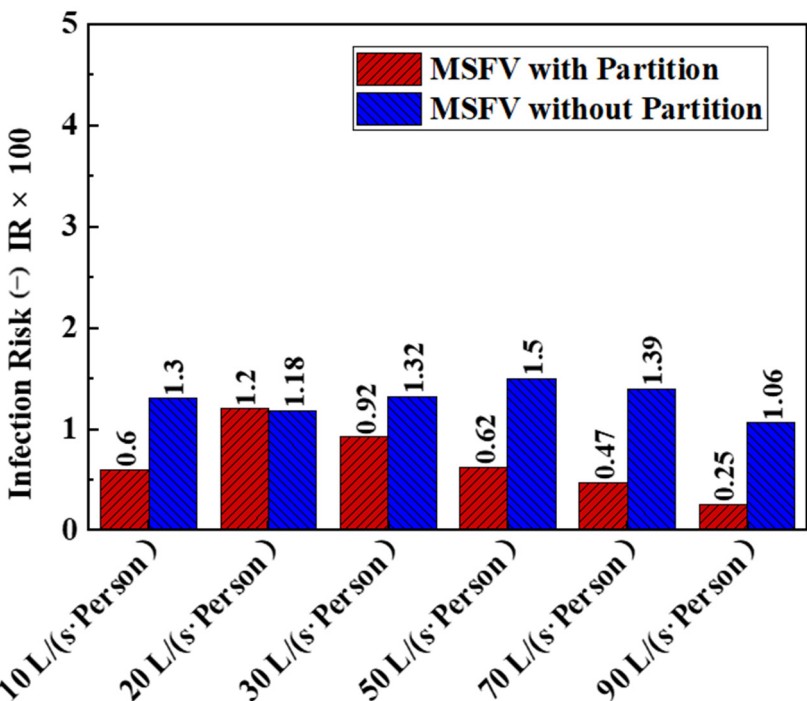

**Figure 17.** Infection risk to medical staff wearing N95 masks under the MSFV, with and without partition.

## 5. Discussion and Conclusions

In this study, we evaluated the exposure risk to medical staff in a nasopharyngeal swab sampling cabin under four different ventilation strategies. Airflow patterns showed an important influence on the concentration distribution of indoor viral substances. Many studies have shown that it is an oversimplification to assume that the air in the space is mixed well. Increasing the ventilation rate may even increase the concentration of the breathing zone in some cases [54–56]. This may be attributed to the complex airflow distribution in the indoor environment caused by the interaction between the ventilation airflow, the thermal plume, and the exhaled airflow. The airflow distribution under the four ventilation strategies may be more complicated because of the partitions. It is reasonable that the exposure risk to medical staff does not decrease linearly with the increase in the ventilation rate. Therefore, in the design stage of the ventilation system, a detailed assessment of the indoor airflow distribution is necessary.

Poor ventilation rate is associated with airborne transmission of the SARS-CoV-2. Some standards or codes [38,39] recommend a minimum ventilation rate for a different type of buildings. However, the ventilation rates were suggested with the goal of diluting bioeffluents (such as odors from people) and achieving basic levels of acceptable indoor air quality, rather than infection control. At present, there is no minimum ventilation rate was recommended based on infection control due to a lack of strong scientific evidence. A few studies may give some advice on the minimum ventilation setting [16,17,57]. For example, Li [16] and Ou [17] reported the outbreaks of COVID-19 in a restaurant and on two buses, respectively. The ventilation rate was measured to be only 1 L/(s·Person) in the restaurant and 1.7 L/(s·Person) and 3.2 L/(s·Person) on the two buses, respectively. Their finding suggests that a ventilation rate of less than 3 L/(s·Person) is not sufficient to prevent airborne transmission of COVID-19 outbreak in the indoor environment. Lu [58] have monitored the $CO_2$ concentration of a hospital during infected person visits to assess the association between ventilation rate and nosocomial infection. They found that the ventilation rate of the throat swab sampling room was 15–58 L/(s·Person), and no medical staff were infected during the contact period (<5 min). Therefore, they recommend a minimum ventilation rate of 15 L/(s·Person). It should be noted that this ventilation rate may be overestimated because the measured minimum ventilation rate is 15 L/(s·Person). The ventilation rate of

10 L/(s·Person) recommended in our study based on energy consumption and infection control may be a suitable one, but more evidence is still needed to verify its rationality.

In the context of the COVID-19 pandemic, the CDC [29] and the WHO [28] have recommended the use of partitions or screening to prevent the spread of infection. The expectation is to add a partition between the infected and the susceptible, which will reduce the physical contact between them and reduce the inhalation of infectious aerosols by the susceptible. The protective effect of the partition on susceptible people may be affected by many factors, such as the position of the exhaust vent [30,31,59], the height of the partition [32], the opening size in the partition [60], etc. For example, Epple [59] found that whether neighbors wear masks or not, partition walls can protect them from bioaerosols. The local exhauster can limit the spread of the exhaled airflow, thereby enhancing the protective effect of the partition. However, Liu [31] reported that the installation of partitions may increase the risk of exposure to surrounding personnel. This may be because the air outlet is far away from the infector, and the pollutants exhaled by the infector cannot be discharged in time and are trapped in the breathing height of the surrounding personnel. Ren [32] confirmed that the installation of partitions will help reduce the exposure level, and a higher partition height means better protection. In our study, the partition is from floor to ceiling, which may help control the spread of pollutants. Nishimura and Sakata [60] have developed a positive/negative pressure booth to protect medical staff from infectious respiratory pathogens when collecting samples. They found that the smaller the opening area, the easier it is to hinder the diffusion of exhaled pollutants. It is necessary to reduce the size of the opening as much as possible if it does not affect the operation of the medical staff.

In this study, only the ventilation system was installed in the sampling chamber. The clean outdoor air was supplied directly into the sampling cabin through the ventilation system without preconditioning. However, the medical staff working in such a thermal environment wearing closed protective clothing may feel uncomfortable. We used the predicted mean vote (PMV) and predicted percent dissatisfaction (PPD) models developed by Fanger [61] to assess the thermal comfort of medical staff. The detailed thermal comfort assessment can be found in the supplementary material. From the distribution of the PMV and PPD, as shown in Figures S3 and S4, it can be that the trend of thermal comfort with ventilation flow was similar to that of exposure risk. The MSFV also performed best on thermal comfort among the four ventilation strategies. The thermal sense of the medical staff was slightly warm and warm, and 30–80% of the medical staff were dissatisfied with the thermal environment. From the medical staff's thermal comfort point of view, it was necessary to precondition the outdoor air before it was fed into the sampling cabin.

This paper numerically investigated the exposure risk to medical staff in a nasopharyngeal swab sampling cabin under different ventilation strategies and ventilation rates. The airflow pattern, normalized concentration at the breathing height in the medical staff zone, the inhaled fraction, and the infection risk to medical staff were evaluated. In addition, the effects of partitions were also discussed. The conclusions drawn include the following:

1.  Among the four ventilation strategies, multiple supply fans ventilation (MSFV) performed best in reducing the exposure risk to medical staff in the sampling cabin in most cases. It is therefore recommended for ventilation of sampling cabins.
2.  The exposure risk to medical staff did not decrease linearly with the increase in the ventilation rate. Considering both the exposure risk to medical staff and the energy consumption of the ventilation, a ventilation rate of 10 L/(s·Person) is recommended for the MSFV.
3.  For the MSFV, installing partitions was beneficial to reduce the risk of exposure in most cases. Under the recommended ventilation flow rate of 10 L/(s·Person), the medical staff's inhaled fraction and exposure risk were reduced by more than 50%.

From our study, both the exposure risk and thermal comfort of the medical staff deserve attention. The comfort of the medical staff can be improved by adjusting the supply air temperature. However, different supply air temperatures may also affect the

diffusion of pollutants exhaled by passengers and further exposure of the medical staff. In addition, the medical staff and passengers may walk in the sampling cabin, which may affect the diffusion and distribution of the exhaled pollutants. Thus, the impact of supply air temperature and personnel walking on the exposure risk to medical staff needs to be further investigated.

**Supplementary Materials:** The following supporting information can be downloaded at: https://www.mdpi.com/article/10.3390/buildings12030353/s1, Figure S1: Model verification by comparing simulation results with experimental results: (a) the model room; (b), (c), (d), and (e) X velocity at x = 0.1, 0.3, 0.5, and 0.7 m; Figure S2: Model verification by comparing simulation results with experimental results: (a) two-person office; (b), (c), and (d) velocity, temperature, and concentration at centerline (x = 2.58 m, z = 1.83 m); Figure S3: The distribution of the PMV; Figure S4: The distribution of the PPD.

**Author Contributions:** Conceptualization, methodology, validation, investigation, software, and data curation, J.M., H.Q. and F.L.; formal analysis, G.S. and X.Z.; resources, J.M. and F.L.; writing—original draft preparation, J.M.; writing—review and editing, all authors; visualization, H.Q. and F.L.; project administration, supervision, H.Q. and X.Z. All authors have read and agreed to the published version of the manuscript.

**Funding:** This research was funded by National Natural Science Foundation of China (Grant No. 41977370), and special fund of Beijing Key Laboratory of indoor Air Quality Evaluation and Control (No. BZ0344KF20-03).

**Institutional Review Board Statement:** Not applicable.

**Informed Consent Statement:** Not applicable.

**Data Availability Statement:** The data used to support the findings of this study are included in the article. The original details of the data presented in this study are available on request from the corresponding author.

**Conflicts of Interest:** The authors declare no conflict of interest.

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
