# Peer review of "Exposure Risk to Medical Staff in a Nasopharyngeal Swab Sampling Cabin under Four Different Ventilation Strategies"

_buildings, doi:10.3390/buildings12030353_

Round 1

Reviewer 1 Report

The article  analyzes numerical exposure to exhaled contaminants by medical personnel performing nasopharyngeal swab tests inside cubicles. The article in its current state cannot be recommended for publication within Buildings. The reviewer finds some problematic points in the paper:
1.    The paper discusses different ventilation strategies for the enclosure under study. Some of the proposed strategies are based on introducing air from outside the room without prior conditioning. From the reviewer's point of view, these are not acceptable ventilation methods.
2.    The study does not take into account the comfort of the occupants of the space under study. The healthcare personnel responsible for the tests in the scenario to be tested have to spend enough time in it so that the conditions of the scenario have to be taken into account with regard to ventilation.
It would be interesting, in order to enrich the document, to consider the comfort values of the occupants of the room at least from the point of view of the healthcare personnel and the people being tested.
3.    The paper presents an analysis of the independence of the results obtained to the mesh used in the simulation. The study focuses on a single point in the room. Due to the characteristics of the room and the different ventilation methods used, it is considered that a more detailed sensitivity analysis considering different ventilation flows and positions within the room would be desirable.
4.    The article presents two different indicators of occupant exposure to pollutants emitted by test persons. It would be interesting to make explicit reference to other articles that present these same indicators in order to be able to compare the results obtained for the case in other scenarios under study.
5.    The results obtained in the article do not allow statements to be made within the conclusions section indicating that a finding of relevance has been obtained through the research carried out.
6.    The document presents some writing problems in the English language. It should be carefully reviewed for possible publication. Some have been indicated that have been found during the review, although there could be others.
 Therefore, it is not possible to recommend the publication of the paper in its present state in the Buildings journal.

Reviewer 2 Report

The work presents exposure risk of medical staff to COVID-19 under the four different ventilation strategies, (i.e., Multiple supply fans ventilation (MSFV), Multiple exhaust fans ventilation (MEFV), Single exhaust fan and keep outer windows close ventilation (SEFV), Single exhaust fan and keep outer windows open ventilation (SEFV-W). The manuscript is interesting and well-prepared, but it deals with ventilation strategy and its effects on the spread of SARS-CoV-2 virus, which has been a frequent topic in articles lately.

The strengths of the work include:
- clear introduction to the issues of work and literature review;
- well-presented research methodology, sample preparation and test execution;
- graphic presentation of the article: photographs, charts, tables, diagrams and drawings which make the manuscript more attractive to the reader;
- detailed analysis of the results and summarizing them in the conclusion.

The weaknesses of the article could be:
- The work is 26 pages long, which may be discouraging for a potential reader. However, it is necessary to reflect the entire spectrum of research and analysis of the work carried out.
- Despite the relatively large size of the manuscript, some figures should be enlarged to make them more legible.

The manuscript needs some corrections before publication: 

  1. It is not good practice to include multiple citations in one sentence without elaborating on them.
    Example - lines 37-39: "SARS-CoV-2 was considered to be transmitted via the virus-laden aerosols emitted by human respiratory activities [4,5], such as breathing, talking, coughing, and sneezing [6-10]." 
  2. Figure 1.
    The medical staff in Figure 1 is described with a yellow figure, and in the description at the bottom of the figure - blue. Please correct it.
  3. Line 197-198.
    Incorrect equation numbering. 
  4. Table 2 and table 3.
    If the table header specifies the data unit "Inhaled Fraction of medical staffs IF × 100%", it can be omitted for reporting the results.

  5. I propose to combine the "Discussion" and "Conclusions" sections into one common section - "Discussion and Conclusions" .
  6. Literature
    According to the journal's guidelines, the literature should be described, for example, "Author 1, T. The title of the cited article. Journal Abbreviation 2008, 10, 142–149."
    After the author's name, a comma should be used and then the initials of the names.
  7. It is worth providing future research directions in the summary of the manuscript. 

Round 2

Reviewer 1 Report

The authors have made the necessary changes for the article to be published.